# Protein Signatures and Individual Circulating Proteins, including IL-6 and IL-15, Associated with Prognosis in Patients with Biliary Tract Cancer

**DOI:** 10.3390/cancers15041062

**Published:** 2023-02-07

**Authors:** Troels D. Christensen, Kasper Madsen, Emil Maag, Ole Larsen, Lars Henrik Jensen, Carsten P. Hansen, Alice Markussen, Dan T. S. Høgdall, Inna M. Chen, Dorte Nielsen, Julia S. Johansen

**Affiliations:** 1Department of Oncology, Copenhagen University Hospital—Herlev and Gentofte, 2730 Herlev, Denmark; 2BioXpedia, 8200 Aarhus, Denmark; 3Department of Oncology, University Hospital of Southern Denmark, 7100 Vejle, Denmark; 4Department of Surgery, Copenhagen University Hospital—Rigshospitalet, 2200 Copenhagen, Denmark; 5Department of Clinical Medicine, Faculty of Health and Medical Sciences, University of Copenhagen, 1165 Copenhagen, Denmark; 6Department of Medicine, Copenhagen University Hospital—Herlev and Gentofte, 2730 Herlev, Denmark

**Keywords:** biliary tract cancer, blood protein assay, cholangiocarcinoma, gallbladder cancer, interleukin, prognostic multimarker panel, survival prediction

## Abstract

**Simple Summary:**

New biomarkers are needed for biliary tract cancer. Here, we tested the prognostic value of 89 circulating proteins. Protein levels were measured in blood samples from four groups of patients treated for biliary tract cancer with either chemotherapy or surgery. Using data from the first patient group, we tested if there was an association between survival and each of the proteins. We also tried to combine the results of several proteins into protein signatures. Afterwards, we validated our observation using data from the three other groups of patients. We identified an association between survival and several proteins, including interleukin 15, which has not been described before. The study also showed that protein signatures, combined with the results of several proteins, were better than individual proteins at identifying patients with short survival.

**Abstract:**

Biliary tract cancer (BTC) is a rare gastrointestinal cancer with a dismal prognosis. Biomarkers with clinical utility are needed. In this study, we investigated the association between survival and 89 immuno-oncology-related proteins, with the aim of identifying prognostic biomarkers for BTC. The study included patients with BTC (*n* = 394) treated at three Danish hospitals. Patients were divided into four cohorts: the first-line discovery cohort (*n* = 202), first-line validation cohort (*n* = 118), second-line cohort (*n* = 56), and surgery cohort (*n* = 41). Plasma protein levels were measured using a proximity extension assay (Olink Proteomics). Twenty-seven proteins were associated with overall survival (OS) in a multivariate analysis in the discovery cohort. In the first-line validation cohort, high levels of interleukin (IL)-6, IL-15, mucin 16, hepatocyte growth factor, programmed cell death ligand 1, and placental growth factor were significantly associated with poor OS in univariate Cox regression analyses. When adjusting for performance status, location, and stage, the association was significant only for IL-6 (hazard ratio (HR) = 1.25, 95% confidence interval (CI) 1.08–1.46) and IL-15 (HR = 2.23, 95% CI 1.48–3.35). Receiver operating characteristic analyses confirmed IL-6 and IL-15 as the strongest predictors of survival. Combining several proteins into signatures further improved the ability to distinguish between patients with short (<6 months) and long survival (>18 months). The study identified several circulating proteins as prognostic biomarkers in patients, with BTC, IL-6, and IL-15 being the most promising markers. Combining proteins in a prognostic signature improved prognostic performance, but future studies are needed to determine the optimal combination and thresholds.

## 1. Introduction

Biliary tract cancers (BTCs) are a group of rare gastrointestinal cancers, including gallbladder cancer (GBC), intrahepatic cholangiocarcinoma (iCC), and extrahepatic cholangiocarcinoma (eCC). The last can be further divided into distal cholangiocarcinoma (dCC) and perihilar cholangiocarcinoma (pCC). The incidence of BTC is low in most countries (about 2–4 cases pr. 100,000). However, regional differences can be seen. In parts of Southeast Asia, the incidence of CC is above 10 pr. 100,000, whereas GBC is common in areas of Chile and Bolivia [1,2,3]. The prognosis is poor and BTC is among the deadliest of malignancies. The median overall survival (OS) is less than a year, and there is an estimated 5-year survival rate of about 20% for all stages [4,5,6].

The poor survival rate is due to late diagnoses and, until recently, limited treatment options in advanced stages. Surgery is the only potentially curative treatment. The 5-year survival rate after surgery ranges between 20–80% depending on tumor location and stage. However, this is only possible for approximately one-third of patients, and recurrence rates are high [3,6,7]. Palliative chemotherapy has been the only recommended treatment option in patients with advanced disease with good performance status (PS) and limited comorbidities. Patients treated with cisplatin and gemcitabine as a first-line therapy are expected to have a median OS of about 12 months based on a phase III randomized clinical trial [8]. Recently, several new treatments have been introduced for patients with BTC. Data have shown improved survival when durvalumab is combined with cisplatin and gemcitabine, and targeted treatments have shown efficacy in smaller subgroups of patients [9,10,11,12,13].

Considering the increasing number of available treatments, estimates of patients’ expected survival are becoming more relevant. A prognostic biomarker might guide decisions on how aggressive treatment should be and when to perform tumor sequencing to detect actionable genomic alterations. Evaluation of the clinical efficacy of new treatments is often performed in non-randomized phase I and phase II studies either without a control group or against external control arms [12,13,14,15], increasing the need for the proper stratification of patients. Potentially, prognostic biomarkers can be utilized to correctly stratify patients in trials to avoid false conclusions. They might also elucidate known and unknown disease mechanisms.

Today, carbohydrate antigen 19-9 (CA19-9) is the most widely used blood-based biomarker for BTC. It is used as a prognostic biomarker and to monitor the effect of treatment and disease progression [16]. However, CA19-9 can also be elevated in benign conditions (e.g., cholangitis) or as a result of biliary stenosis, and 10% of BTC patients do not express CA19-9 [16]. Other biomarkers linked to BTC prognosis include interleukin (IL)-6 [17], the systemic inflammation index (SII), C-reactive protein (CRP), and the neutrophile-to-lymphocyte ratio (NLR) [17,18,19], but the prognostic accuracy is relatively low, and novel prognostic biomarkers are still needed.

Circulating proteins besides IL-6 and CRP might be associated with patient survival [20,21,22,23,24]. Notably, proteins related to the tumor microenvironment, inflammation, and cancer immune reaction are of interest to explore in BTC. The desmoplastic tumor microenvironment of BTC is highly interactive, and multitudes of proteins are produced and secreted by cancer and non-cancer cells, such as immune cells and cancer-associated fibroblasts. This complex cross-talk can stimulate tumor growth, invasion, angiogenesis, immunosuppression, and metastatic spread, as well as extracellular matrix modulation and inflammation [2,17,25,26,27]. Several of the proteins released as part of this process leak to the bloodstream, leading to changes in the blood proteome, and many proteins have been found elevated in patients with BTC compared to controls [27,28]. However, very few proteins have been tested as prognostic markers, creating the great potential for new studies.

To discover novel biomarkers, a feasible approach is to use multiplex protein arrays to test several proteins simultaneously in one cohort, followed by validation in an independent cohort. Our group has previously shown that novel prognostic biomarkers could be identified using a protein panel with immuno-oncology (I-O) and inflammation-related proteins in patients with pancreatic cancer [23]. A similar approach might be used to find novel prognostic biomarkers for patients with BTC.

In this study, we investigated the association between the prognoses of patients with BTC and 89 I-O-related proteins, including several inflammation biomarkers. We both evaluated the potential prognostic application of individual proteins and the combination of proteins with biomarker signatures.

## 2. Materials and Methods

### 2.1. Patients

The present study retrospectively included 394 patients with BTC (iCC, eCC, and GBC) enrolled in prospective clinical studies between October 2008 and May 2021 at three Danish hospitals (Herlev Hospital, Vejle Hospital, and Rigshospitalet) (Figure 1). The patients from Herlev Hospital and Vejle Hospital had been referred for first- or second-line oncological treatment for unresectable BTC. These patients had been included and treated in clinical trials (GI1003 [29], GI1333 [30], GI1312 [31], GOC-BP [32], and GOX-P [33,34]) or received standard treatment in the open prospective cohort study CHOCA (NCT05184400). The patients from Rigshospitalet had been referred for tumor resection and included in the open prospective cohort study BIOPAC (NCT03311776).

Inclusion criteria were histological confirmed BTC (iCC, eCC, or GBC) and an available blood sample collected before the intended treatment (surgery, or first-line or second-line oncological treatment). Exclusion criteria were other cancers (except patients with benign cancers of the skin or patients with radically treated cancers and no sign of relapse) at diagnosis or patients diagnosed with secondary cancer within two years. Likewise, patients with a mixed histology (hepatocellular-cholangiocarcinoma) were excluded. Lastly, patients enrolled in ongoing 1st- or 2nd-line treatment protocols (NCT02866383 and EudraCT 2018-004826-27) were excluded due to unavailable follow-up data. Detailed descriptions of each study are available in the Appendix A.

Patients with BTC were divided into four cohorts to explore the prognostic performance of proteins and protein signatures (Figure 1): (1) the first-line discovery cohort consisted of patients with unresectable BTC included from Herlev Hospital (*n* = 195) or Rigshospitalet (*n* = 7) and had blood samples collected before first-line treatment at Herlev Hospital; (2) the first-line validation cohort consisted of patients with unresectable BTC with blood samples collected before first-line treatment at Vejle Hospital (*n* = 118); (3) the second-line cohort consisted of patients with unresectable BTC with blood samples collected before second-line treatment at Herlev Hospital (*n* = 56); and (4) the surgery cohort consisted of patients with resectable BTC with blood samples collected before surgery at Rigshospitalet (*n* = 41). Of note, 23 patients in the discovery cohort also had samples collected before surgery or second-line treatment and were included in both the discovery cohort and one of the other cohorts (4 in the surgery cohort and 19 in the second-line cohort).

### 2.2. Sample Characteristics

Serum samples were used for the analyses in patients with BTC included from Herlev Hospital and Rigshospitalet. Ethylenediaminetetraacetic acid (EDTA) plasma samples were used in patients included from Vejle Hospital.

Samples from BTC patients were collected prior to the initiation of therapy. The serum samples were collected in 8 mL serum separator tubes and centrifuged at 2300× *g* at 4 °C for 10 min. Serum was then aliquoted in Greiner tubes (Cryo.s™ Freezing Tubes, 2 mL; GR-121280, Greiner Bio-One GmbH, Kremsmünster, Austria). The EDTA plasma samples were prepared by centrifuging blood samples collected in 9 mL EDTA tubes at 1486× *g* at 21 °C for 10 min, and plasma was then transferred to 15 mL cryo-tubes. The samples were subsequently stored at −80 °C.

### 2.3. Biomarker Analyses

Blood samples were analyzed using the proximity extension assay (PEA) basedOlink Target 96 Immuno-Oncology panel (Olink Proteomics, Uppsala, Sweden). Protein levels were measured on a relative scale and presented as a normalized protein expression (NPX), which is an arbitrary unit on a log2 scale. A high NPX value corresponds to a high protein concentration [35]. The panel covers proteins previously associated with cancer, the immune system, and systemic inflammation. A full list of proteins is available in Appendix A.

The analyses were performed at BioXpedia, Aarhus, Denmark. BioXpedia was blinded to the study endpoint as no research questions or clinical data were passed on before all the samples had been analyzed. A trained bioinformatician performed normalization and quality control according to the manufacturer’s recommendations (EM). Samples were normalized for any plate effects using the built-in interplate controls and bridging samples. Samples or proteins were removed if they failed quality control (Appendix A). The effect of storage was also investigated to ensure that results were comparable between samples collected at different time points.

Routine laboratory results (albumin, CA19-9, CRP, neutrophils, lymphocytes, and platelet count) measured of the same day as the study sample collection were retrieved from the study databases. When possible, serum samples from patients with missing clinical values were analyzed retrospectively. NLR was calculated as neutrophil count/lymphocyte count. SII was calculated as neutrophil count/lymphocyte count x platelet count.

A thorough description of biomarker analyses is available in the Appendix A.

### 2.4. Statistics

Results are reported in accordance with the REMARK (reporting recommendations for tumor marker prognostic study guidelines) [36] (Appendix A). This is an exploratory study that included all available patients, and thus, no sample size was calculated.

Protein levels were compared between groups of interest using a *t* test if there were more than 30 patients in each group or data were normally distributed, which was determined using the Shapiro–Wilk test. Otherwise, the Wilcoxon rank-sum test was applied. The *p*-values were corrected for multiple testing using the Benjamini–Hochberg method. The log2 fold change was calculated on a linear scale using the geometric mean of each group.

The correlation between markers was examined using Pearson’s correlation.

Associations between survival and individual protein levels were visualized using Kaplan–Meier plots. Cox regression analyses were performed using data from the first-line discovery cohort. All biomarkers were analyzed as continuous variables. The endpoint was OS from date of blood sampling to death by any cause or last follow-up (7 February 2022). First, univariate Cox regression analysis was performed for all proteins. The 10 proteins with the lowest *p*-value in the univariate analysis were analyzed with the multivariate Cox regression, adjusting for PS (0–1 vs. 2–4), stage (locally advanced vs. metastatic), location (iCC vs. extrahepatic (GBC, pCC, dCC)), CA19-9 (kU/L), and age (years). Significant variables were tested in the validation cohorts using both univariate and multivariate Cox regression adjusting for PS and location after removing patients who had also been included in the first-line discovery cohort.

Protein signatures were generated using ridge and lasso regression (see Appendix A). The performance of individual proteins and signatures at distinguishing between patients with short and long survival (OS ≤ 12 months vs. OS > 12 months, OS ≤ 6 months vs. OS > 18 months, and OS ≤ 3 months vs. OS > 24 months) was evaluated in each cohort using receiver operating characteristic (ROC) curves and area under the ROC curve (AUC). The optimal cut-off was calculated using Youden’s index [37].

Statistical analyses were performed by a trained bioinformatician (EM) and a statistician (KM) in collaboration with medical doctors following a prespecified analysis plan. All analyses were performed using R version 4.1.2 and 4.2.0 [38]. A two-sided *p*-value of 0.05 was considered significant.

## 3. Results

Of the 394 patients enrolled in the study, 7 patients were excluded after sample quality control, and only 387 patients were used in further analyses. A total of 198 patients were included in the first-line discovery cohort, 117 in the first-line validation cohort, 54 in the second-line cohort, and 40 in the surgery cohort (Figure 1).

Twenty-one patients were alive at last follow-up on 7 February 2022, and one patient (first-line discovery cohort) had a follow-up of <1 year (263 days). The baseline characteristics are shown in Table 1.

A total of 89 proteins were used in the analyses. Three proteins (IL-1 alpha, IL-2, and IL-13) were removed before further analysis because more than 90% of samples had values below the limit of detection across all cohorts. Protein levels between samples collected at different time points were compatible for all proteins (Appendix A).

### 3.1. Survival Analyses

In the discovery cohort, CA19-9 and 44 of the 89 tested I-O proteins had a significant association with OS in univariate analyses after adjusting for multiple testing. For all proteins, increased plasma levels were associated with shorter OS. The association remained significant for 29 of the 44 proteins in multivariate analyses adjusting for CA19-9, PS, location of primary tumor, sex, age, and stage stratification (Appendix A). The 10 proteins with the lowest *p*-values were IL-15, IL-6, mucin 16 (MUC-16), programmed cell death ligand (PD-L) 1, PD-L2, natural cytotoxicity triggering receptor (NCR1), hepatocyte growth factor (HGF), endothelial nitric oxide synthase (NOS3), C-C motif chemokine (CCL) 20, and placental growth factor (PGF).

A significant difference was observed in OS between the lowest tertile and highest tertile for all these proteins on Kaplan–Meier plots (Figure 2A). Likewise, median OS was markedly longer in patients with a protein level in the lowest tertile compared with the highest tertile. As an example, in the discovery cohort, the median OS for patients with IL-15 in the highest tertile was 144 days vs. 531 days for patients with IL-15 in the lowest tertile. A similar difference was seen for IL-6 (158 days vs. 534 days). A subgroup analyses of patients receiving gemcitabine plus cisplatin or gemcitabine plus capecitabine plus oxaliplatin combinations revealed an association with OS similar to that in the analyses performed for the entire cohort.

The top 10 proteins were selected for validation in the three other cohorts (Figure 2B, Appendix A). In the first-line validation cohort, increased levels of IL-15, IL-6, MUC-16, HGF, PD-L1, and PGF were associated with worse survival. Patients with protein levels in the highest tertile had a shorter survival than patients with protein levels in the lowest tertile (Appendix A). For IL-15, the median OS for patients within the highest tertile was 205 days compared to 368 days for patients with levels in the lowest tertial. A similar difference was observed for IL-6 (205 days vs. 399 days). When adjusting for PS, tumor location, and stage, only IL-6 (HR = 1.25, 95% confidence interval (CI) 1.08–1.46) and IL-15 (HR = 2.23, 95% CI 1.48–3.35) remained significantly associated with OS.

In the second-line cohort, increased IL-15, IL-6, PD-L1, PGF, CCL20, and MUC-16 were associated with shorter OS in the univariate and multivariate analyses. However, after adjusting for multiple comparisons, no markers remained significantly associated with OS in the second-line cohort. In the surgical cohort, none of the 10 proteins were associated with OS.

### 3.2. Comparing Biomarker Levels in Patients with Short and Long Survival in the Discovery Cohort

Thirty-six proteins were identified as significantly elevated in patients with short survival when all BTC patients with an OS ≤ 12 months vs. an OS > 12 months were compared. All but one (IL-7) was also significantly increased in patients with short survival in the comparison between patients with an OS ≤ 6 months vs. an OS > 18 months, and an OS ≤ 3 months vs. an OS > 24 months (Appendix A). All 10 markers (IL-15, IL-6, MUC-16, PD-L1, PD-L2, NCR1, HGF, NOS3, CCL20, and PGF) evaluated in the validation analyses were significantly elevated across the three comparisons. When comparing protein levels between short- and long-term survivors in subgroups according to treatment (gemcitabine plus capecitabine vs. gemcitabine plus capecitabine plus oxaliplatin combination) or tumor location, a similar pattern was detected. However, IL-6, IL-15, and MUC-16 were not significantly elevated among short-term-surviving patients with GBC. Appendix A shows all comparisons for all groups.

ROC and AUC analyses confirmed the ability of IL-6 and IL-15 to discriminate between short- and long-term survivors (Figure 3). In the first-line discovery cohort, the AUC was 0.75 (IL-6) and 0.77 (IL-15) for an OS ≤ 12 months vs. an OS > 12 months, 0.87 for an OS ≤ 6 months vs. an OS > 18 months (similar for IL-6 and IL-15) and 0.88 (IL-6) and 0.89 (IL-15) for an OS ≤ 3 months vs. an OS > 24 months. Across the three comparisons, MUC-16, PD-L1, PD-L2, HGF, CCL23, macrophage colony-stimulating factor pleiotrophin (CSF-1), tumor necrosis factor receptor superfamily member 12A (TNFRSF12A), and PGF were also among the 10 best performing proteins at discriminating between survival groups with AUCs above 0.69, 0.77, and 0.75. The top 10 best performing proteins in the first-line discovery cohort and CA19-9 were tested in the validation cohort. In the first-line validation cohort, IL-6 and IL-15 remained among the best performing proteins, with an AUC of 0.64 (IL-6) and 0.66 (IL-15) (OS ≤ 12 months vs. OS > 12 months), 0.78 (IL-6) and 0.79 (IL-15) (OS ≤ 6 months vs. OS > 18 months), and 0.77 (IL-6 and IL-15) (OS ≤ 3 months vs. OS > 24 months). CA19-9 had AUCs of 0.60, 0.70, and 0.50 for the three comparisons. In the second-line and surgical cohorts, IL-6 and IL-15 showed reasonable performance, but the AUC was not superior to several other markers across all comparisons. Appendix A shows the AUC, confidence interval, best point, sensitivity, and specificity for all comparisons.

### 3.3. Association and Performance Compared to Other Inflammation and Prognostic Factors

Using available data from the first-line discovery cohort, the performance of IL-6 and IL-15 was compared with CA19-9, CRP, neutrophil count, lymphocyte count, NLR, SII, albumin, and bilirubin. IL-15 and IL-6 performed better than all these markers. Of the other markers, SII, NLR, and CRP had the highest AUCs (Figure 4, Appendix A). We only had data regarding CA19-9 levels in the three other cohorts. In the first-line validation cohort, IL-15 and IL-6 performed better than CA19-9. However, in the second-line cohort and surgery cohort, they showed a performance similar to that of CA19-9 across comparisons (Figure 4).

The correlation between known prognostic biomarkers and selected potential prognostic proteins was also evaluated (Appendix A). The correlation analyses identified a strong correlation (*r* > 0.6) between several of the prognostic proteins (CCL23, IL-15, HGF, NCR1, PD-L1, PD-L2, PGF, and TNFRSF12A). Most of these biomarkers had weaker correlations (*r* < 0.4) with other inflammation biomarkers such as CRP or albumin. IL-6 showed a moderate correlation with most of the other I-O protein biomarkers, albumin, neutrophil count, and CRP (*r* 0.4–0.6). A weak or no association was observed between most I-O proteins and lymphocytes, NLR, SII, CA19-9, and liver markers (*r* < 0.4). Of note, most patients had a bilirubin level below 25 µmol/L, and only seven had a bilirubin level above 50 µmol/L, hampering tests for the effect of elevated bilirubin.

Both IL-15 and IL-6 were elevated in patients with metastatic disease compared to patients with non-metastatic disease, as was identified via comparing levels in the first available serum sample across cohorts. However, the difference between resectable disease and metastatic disease was not significant for IL-15. Interestingly, IL-15 was also higher in the second-line cohort compared to the first-line cohort and surgery cohort (Appendix A).

### 3.4. Protein Signatures

The multivariate logistic lasso regression including all 89 proteins, and CA19-9 confirmed that IL-6 and IL-15 were two of the top 10 predictors at discriminating between short- and long-term survivors. Three other proteins (MUC-16, CCL23, and PTN) were also among the top 10 proteins (Appendix A).

Three sets of signatures were generated using lasso and ridge regression (Appendix A). The signatures achieved high AUCs in the first-line discovery cohort (detection and replication set) and all validation cohorts (Table 2, Appendix A). However, the best performing signatures had an AUC similar to the best performing proteins (IL-6 and IL-15) for comparing patients with an OS ≤ 12 months vs. an OS > 12 months (Set 1). Thus, a signature combining IL-6, MUC-16, IL-10, PTN, and CA19-9 achieved an AUC of 0.75 in the replication set of the first-line discovery cohort and 0.66 in the first-line validation cohort, and 0.68 in the surgical cohort and 0.69 in the second-line cohort.

The signatures generated to distinguish between patients with an OS ≤ 6 months vs. an OS > 18 months achieved a higher AUC than IL-6 and IL-15 in both replication and all three validation cohorts. Thus, a signature combining IL-6, MUC-16, IL-15, CCL23, MHC class I polypeptide-related sequence A/B (MICA/B), and PTN achieved an AUC of 0.91 in the replication set, 0.82 in the first-line validation cohort, 0.85 in the surgical cohort, and 0.87 in the second-line cohort (Table 2, Appendix A). Similarly, signatures generated to distinguish between patients with an OS ≤ 3 months vs. an OS > 24 months (Set 3) performed better than individual proteins in this comparison. Only minor differences in the ability to discriminate between short- and long-term survivors were seen between signatures, and no clear trend of improved discriminatory ability was observed by combining more than six proteins in one signature (Table 2, Appendix A).

## 4. Discussion

Our study investigated the prognostic use of 89 circulating proteins in patients with BTC using a discovery cohort and independent validation cohorts. The study explored the prognostic potential of individual proteins and combinations of proteins in prognostic signatures. The strongest and most consistent association for individual proteins was seen between a short OS and high IL-15 and IL-6 levels. These two proteins outperformed known prognostic biomarkers such as CRP, NLR, and SII. Four other markers (PD-L1, MUC-16, HGF, and PGF) were also associated with OS, although the results were not significant in validation studies after adjusting for other factors and multiple testing. Our study is the first to investigate and identify an association between high levels of circulating IL-15 and short survival in patients with BTC. We also identified several multiprotein signatures that performed better than individual proteins.

In the neoplastic setting, IL-15 is likely produced by myeloid cells of the tumor microenvironment and cancer cells [39]. IL-15 exists in a membrane-bound and soluble form, and the soluble form can exist both unbound and in complex with the IL-15 α-helix receptor (IL-15Rα) [39]. Previously, only a few small studies had examined the association between the clinical outcome in patients with cancer and the level of IL-15 or its receptors (IL-15R) [40,41]. IL-15 has been shown to have a pro-tumorigenic role and is related to tumor escape, aggressive tumor behavior, and proinflammatory signaling [39]. A high intratumor expression of IL-15 and serum IL-15R has been associated with poor survival in lung cancer [42] and head and neck cancers, respectively [41]. However, IL-15 might also have an antineoplastic role and supports the development and function of natural killer cells and CD8+ T-cells. High IL-15 expression is linked to a decreased recurrence in patients with colorectal cancer [39,43], and an increased level of plasma IL-15 was associated with clinical benefit of checkpoint inhibitors in lung cancer patients [40]. This potential antineoplastic effect has led to the development of an IL-15 superagonist [44], which has shown clinical efficacy in phase I and II studies in lung and bladder cancer patients [45,46].

In this regard, our results are interesting since they indicate that IL-15 might have a detrimental effect in patients with BTC. However, a causal effect was not established in our study. Besides the potential direct pro-tumorigenic effect, the association may reflect increased liver damage and inflammation in patients with a high IL-15 level, as previously observed in nonalcoholic steatohepatitis [47]. Unfortunately, we had no available data to test this hypothesis, and future studies should investigate if this explains our observations.

The strong association between a high plasma IL-6 level and poor survival has previously been found in a subset of the investigated patient cohorts using an enzyme-linked immunosorbent assay to determine the protein [17]. IL-6 plays a central role in chronic inflammation. It stimulates cancer growth and prevents apoptosis, suggesting that IL-6 is a potential treatment target [17,25]. Cell and mouse studies have indicated the potential use of the anti-IL-6 receptor antibody tocilizumab for treatment of various cancers including BTC [17,48]. An ongoing randomized phase 2 study is testing gemcitabine plus cisplatin with or without tocilizumab in patients with BTC (EudraCT number 2018-004826-27).

Circulating PD-L1, HGF, and MUC-16 have been associated with poor survival in BTC and other cancers [20,21,22,23,24], but all previous studies on BTC included less than 250 patients and lacked independent validation cohorts. Besides a small study including 19 patients [49], no studies have previously investigated the prognostic value of circulating PGF in patients with BTC. However, an increased level of circulating PGF is associated with poor survival in patients with pancreatic cancer [23], and PGF expression in tumor tissue has been linked to CC tumor growth in tumor models [50]. The available data together with our results indicate that these proteins (especially HGF, MUC-16, and PD-L1) could have a prognostic use in patients with BTC, but more studies are needed to validate this.

The protein signatures identified in the present study varied between the three survival comparisons, even though IL-6, IL-15, PTN, CCL23, and MUC-16 were used in most signatures. Including more than six proteins in the signatures did not add extra information. The reason for this is probably that the levels of many proteins are highly correlated, and adding extra proteins gives more noise than information. The high correlation between these protein biomarkers could also explain why PD-L1, HGF, and PGF were only included in some of the larger signatures with more than 12 proteins. In our previous study on pancreatic cancer, protein signatures that included four and seven proteins achieved AUCs of 0.82 and 0.89 for distinguishing between patients with short and long survival (<6 months vs. >18 months or <3 months vs. >24 months). Of the five most used proteins in the BTC signatures (IL-6, IL-15, PTN, CCL23, and MUC-16), only IL-6 was used in the primary pancreatic cancer signatures. This study did not include an external validation cohort. Therefore, results are best compared with our replication set in the discovery cohort, and here, the BTC signatures showed a similar performance.

The use of the circulating proteins investigated here in patients with BTC is not limited to prognostic applications. Of the 44 proteins significantly associated with OS in the discovery cohort, 39 have been shown to have elevated plasma levels in patients with BTC as compared to controls (Appendix A) [28]. The six proteins most prominently associated with OS (IL-6, IL-15, PD-L1, MUC-16, HGF, and PGF) were all elevated in patients with BTC as compared with controls (Appendix A and submitted data). Our previous study also identified a potential diagnostic role of multiprotein signatures [28].

The main strength of the study is the multiple cohort design which allowed for independent validation of the results. Two different blood matrices were used in the discovery cohort (serum) and first-line validation cohort (EDTA plasma). This can affect the protein level measured using the PEA platform [51,52]. Therefore, it is also a strength of the study that we were able to replicate our results across blood matrices. However, this might have led to an underestimation of the protein signatures’ performance in the first-line validation cohort.

The present study has some limitations. Although the study is larger than most biomarker studies in patients with BTC, our validation cohorts are underpowered to detect association with survival. The lack of significance in multivariate analyses in the validation cohorts should, therefore, be interpreted with some caution. Notably, the second-line cohort and surgery cohorts were too small for firm conclusions. Likewise, our subgroup analyses in the first-line discovery cohort were based on a small number of patients, and the results have a high degree of uncertainty. Moreover, the PEA technology used in the study measures protein level as relative abundance. This does not affect the validity of the association observed with OS but limits the ability to establish cut-off values for use in future studies. Furthermore, we did not have information regarding mutational status, and how this impacts the association between protein level and survival is also unknown. We hope to test this using data from an ongoing trial (EudraCT 2018-004826-27). Finally, besides our small cohort of surgically resected patients, almost all other patients received chemotherapy. Patients received different combinations of chemotherapy, and the biomarkers seem to be prognostically independent of treatment regimen. However, we cannot completely rule out that some biomarkers are specifically predictive of chemotherapy efficacy. A different result would potentially have been seen in a cohort of patients treated with immunotherapy given the association between several of the proteins and the immune system. We expect to test this in plasma samples in an ongoing trial combining immunotherapy and radiation therapy for patients with metastatic BTC (NCT02866383).

## 5. Conclusions

Out of 89 circulating I-O-related proteins, our study identified IL-6 and IL-15 as the most robust individual prognostic biomarkers in patients with BTC. Both outperformed CA19-9, CRP, NLR, and SII. The association was validated for IL-6 (HR = 1.25, 95% CI 1.08–1.46) and IL-15 (HR = 2.23, 95% CI 1.48–3.35) in an independent cohort adjusting for PS, tumor location, and stage. Other proteins, including HGF, PD-L1, PGF, and MUC-16, were also associated with survival. We also generated and tested several protein signatures (most including IL-6 and IL-15). The protein signatures showed a stronger ability to distinguish between patients with short (<6 months) and long survival (>18 months) than individual markers, but future studies are needed to determine the best combination.

## Figures and Tables

**Figure 1 cancers-15-01062-f001:**
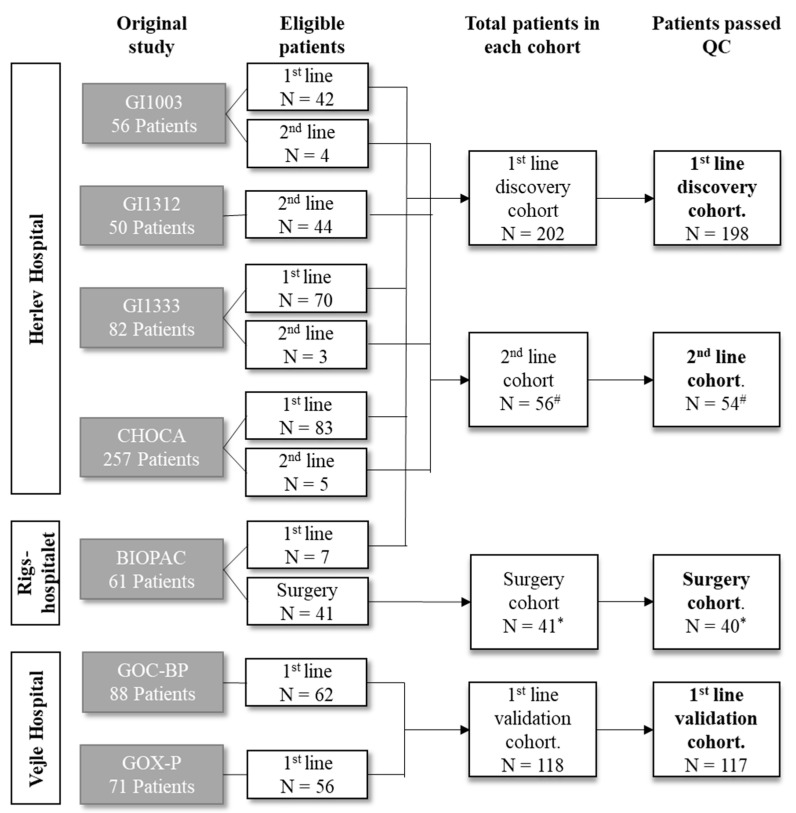
Cohort diagram. Abbreviation: *n*—number of patients; QC—sample quality control. ^#^ Nineteen patients before QC and eighteen after QC were also included in the first-line cohort. * Four patients were also included in first-line cohort.

**Figure 2 cancers-15-01062-f002:**
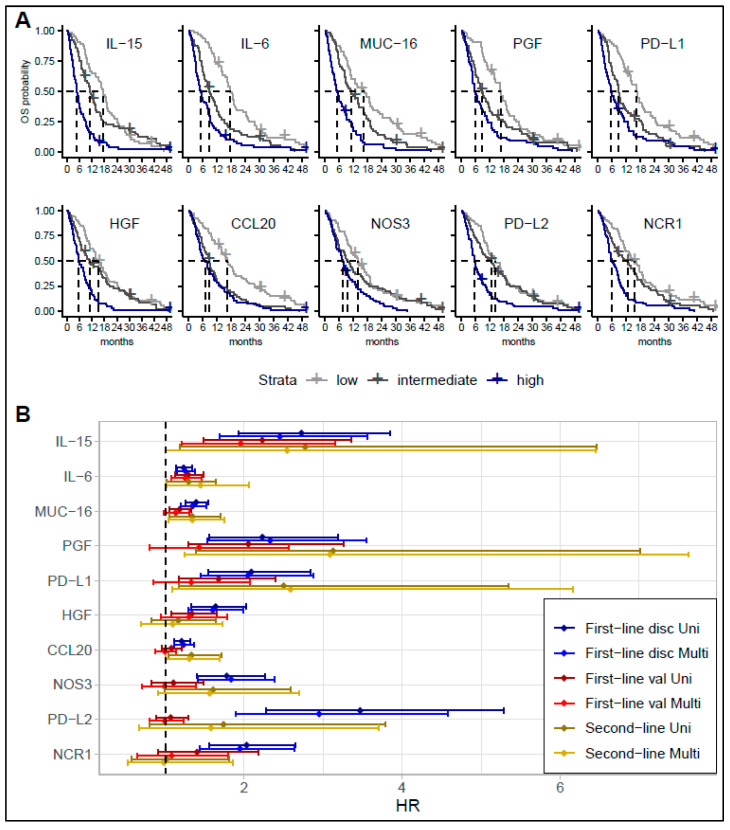
Association between overall survival (OS) and biomarker level. Plot shows results of survival analysis for the proteins with the strongest association with OS. (**A**): Kaplan–Meier curves showing survival probability in first-line discovery cohort according to biomarker level divided into tertiles (low, intermediate, or high). (**B**): Forrest plot showing results of univariate and multivariate Cox regression analysis in first-line discovery cohort, first-line validation cohort, and second-line cohort. Multivariate analysis adjusting for carbohydrate antigen (CA)19-9, performance status (PS), location of primary tumor, sex, age, and stage stratification in discovery cohort, and fir PS, location of primary tumor, and stage stratification in both validation cohorts. Abbreviations: disc—discovery cohort; CCL20—C-C motif chemokine 20; HGF—hepatocyte growth factor; IL—interleukin; NCR1—natural cytotoxicity triggering receptor; NOS3—endothelial nitric oxide synthase; MUC-16—mucin 16; Multi—multivariate analyses; PD-L1—programmed cell death 1 ligand 1; PD-L2—programmed cell death 1 ligand 2; PGF—placental growth factor; val—validation cohort; Uni—univariate analyses.

**Figure 3 cancers-15-01062-f003:**
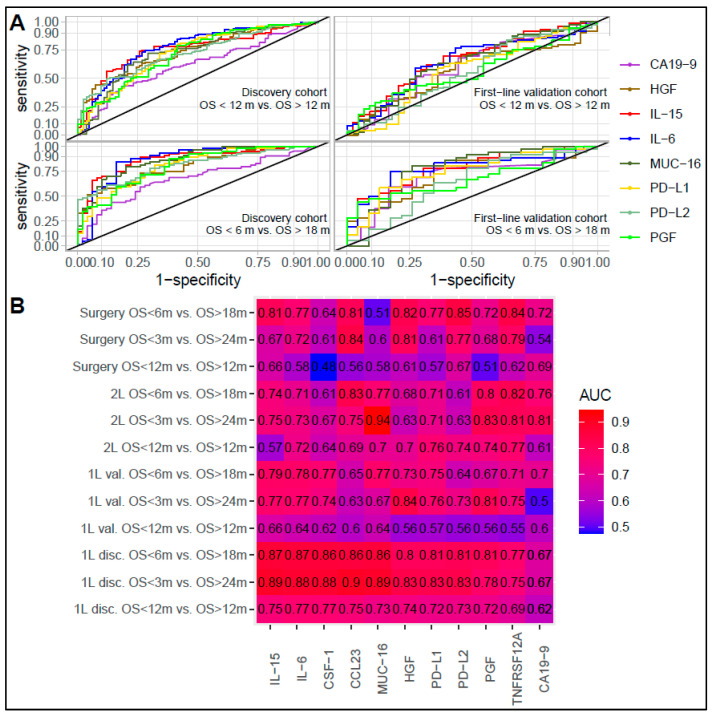
Ability of individual proteins to discriminate between patients with short or long survival. (**A**): Receiver operating characteristic (ROC) curve for best performing proteins and CA19-9. Plot shows ability to discriminate between patients with short and long survival in discovery cohort and first-line validation cohort. (**B**): Area under the ROC curve (AUC) for 10 proteins with highest AUC in discovery cohort and carbohydrate antigen (CA) 19-9 for all comparisons in all cohorts. Abbreviations: AUC—area under the curve; 1L disc.—first-line discovery cohort; 1L val.—first-line validation cohort; 2L—second-line cohort; CA19-9—carbohydrate antigen 19-9; CCL23—C-C motif chemokine 23; CSF-1—macrophage colony-stimulating factor 1; HGF—hepatocyte growth factor; IL—interleukin; m—months; MUC-16—mucin 16; OS—overall survival; PD-L1—programmed cell death 1 ligand 1; PD-L2—programmed cell death 1 ligand 2; PGF—placental growth factor; TNFRSF—tumor necrosis factor receptor superfamily member.

**Figure 4 cancers-15-01062-f004:**
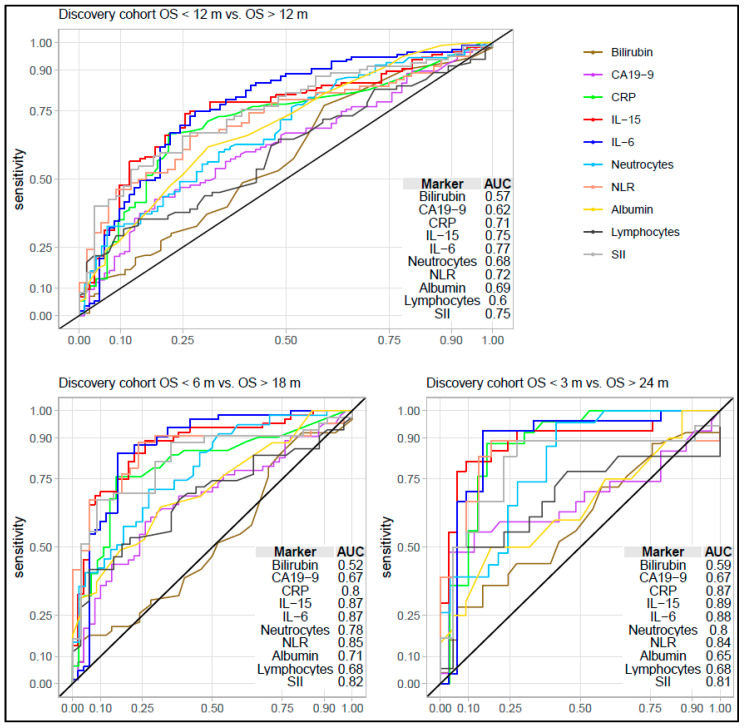
Performance of interleukin (IL)-15 and IL-6 compared to other prognostic markers in the first-line discovery cohort. Receiver operating characteristic curve (ROC) and area under the ROC curve (AUC) for IL-15 and IL-16 compared to routine prognostic clinical biomarkers of inflammation. Abbreviations: CA19-9—carbohydrate antigen 19-9; CRP—c-reactive protein; IL—interleukin; m—months; NLR—neutrocyte-to-lymphocyte ratio (neutrocytes/lymphocytes); OS—overall survival; SII—systemic inflammation index (platelets x neutrocytes/lymphocytes).

**Table 1 cancers-15-01062-t001:** Patient characteristics in the four cohorts of patients with biliary tract cancer after removing patients whose sample failed quality control.

		First-Line Discovery Cohort	First-Line Validation Cohort	Surgery Cohort	Second-Line Cohort
No. of patients after removing samples that failed quality control		198	117	40	54
Sex (%)	Female	115 (58.1)	75 (64.1)	19 (47.5)	31 (57.4)
	Male	83 (41.9)	42 (35.9)	21 (52.5)	23 (42.6)
Age—years (median [IQR])		67 (59, 72)	66 (59, 73)	67 (56, 70)	67 (59, 71)
Performance status (%)	0	106 (53.5)	41 (35.0)	16 (40.0)	32 (59.3)
	1	78 (39.4)	52 (44.4)	12 (30.0)	20 (37.0)
	2	10 (5.1)	24 (20.5)	1 (2.5)	2 (3.7)
	Unknown	4 (2.0)	0 (0.0)	11 (27.5)	0 (0.0)
Stage (%)	Resectable	0 (0.0)	0 (0.0)	40 (100.0)	0 (0.0)
	Locally advanced	87 (43.9)	24 (20.5)	0 (0.0)	6 (11.1)
	Metastatic	111 (56.1)	93 (79.5)	0 (0.0)	48 (88.9)
Location (%)	iCC	108 (54.5)	46 (39.3)	0 (0.0)	29 (53.7)
	pCC	28 (14.1)	20 (17.1)	0 (0.0)	10 (18.5)
	GBC	40 (20.2)	23 (19.7)	1 (2.5)	13 (24.1)
	dCC	22 (11.1)	19 (16.2)	39 (97.5)	2 (3.7)
	Unknown	0 (0.0)	9 (7.7)	0 (0.0)	0 (0.0)
History of liver disease *	No	190 (96.0)	0 (0.0)	40 (100.0)	45 (83.3)
	Yes	8 (4.0)	0 (0.0)	0 (0.0)	4 (7.4)
	Unknown	0 (0.0)	117 (100.0)	0 (0.0)	5 (9.3)
Resection of primary tumor	No	169 (85.4)	103 (88.0)	0 (0.0)	39 (72.2)
	Yes	29 (14.6)	14 (12.0)	40 (100.0)	14 (25.9)
	Unknown	0 (0.0)	0 (0.0)	0 (0.0)	1 (1.9)
Adjuvant chemotherapy	No	178 (89.9)	0 (0.0)	9 (22.5)	48 (88.9)
	Yes	20 (10.1)	0 (0.0)	31 (77.5)	6 (11.1)
	Unknown	0 (0.0)	117 (100.0)	0 (0.0)	0 (0.0)
First-line treatment	No	8 (4.0)	5 (4.3)	9 (22.5)	0 (0.0)
	Gem	11 (5.6)	0 (0.0)	1 (2.5)	0 (0.0)
	Other	7 (3.5)	0 (0.0)	8 (20.0)	1 (1.9)
	GemCapOx	48 (24.2)	14 (12.0)	3 (7.5)	37 (68.5)
	GemCapOx + mAB	44 (22.2)	98 (83.8)	0 (0.0)	7 (13.0)
	GemCis	80 (40.4)	0 (0.0)	4 (10.0)	8 (14.8)
	Unknown	0 (0.0)	0 (0.0)	15 (37.5)	1 (1.9)
Second-line treatment	None	119 (60.1)	0 (0.0)	34 (85.0)	4 (7.4)
	GemCapIri	25 (12.6)	0 (0.0)	0 (0.0)	4 (7.4)
	GemCapIri + mAB	21 (10.6)	0 (0.0)	1 (2.5)	42 (77.8)
	Other	33 (16.7)	0 (0.0)	5 (12.5)	4 (7.4)
	Unknown	0 (0.0)	117 (100.0)	0 (0.0)	0 (0.0)
OS status	Alive	8 (4.0)	4 (3.4)	9 (22.5)	0 (0.0)
	Dead	190 (96.0)	113 (96.6)	31 (77.5)	54 (100.0)
OS	<90 days	27 (13.6)	17 (14.5)	3 (7.5)	13 (24.1)
	90–179 days	37 (18.7)	19 (16.2)	2 (5.0)	19 (35.2)
	180–364 days	52 (26.3)	37 (31.6)	7 (17.5)	8 (14.8)
	365–547 days	32 (16.2)	21 (17.9)	4 (10.0)	8 (14.8)
	548–729 days	17 (8.6)	11 (9.4)	5 (12.5)	2 (3.7)
	>730 days	33 (16.7)	12 (10.3)	19 (47.5)	4 (7.4)
CA19-9, kU/L (median [IQR])		177 (56, 1918)	161 (39, 1823)	135 (29, 330)	171 (39, 749)
ALAT, U/L (median [IQR])		41 (28, 66)	NA	60 (34, 123)	48 (32, 119)
ASAT U/L (median [IQR])		52 (37, 84)	NA	NA	81 (74, 133)
ALP U/L (median [IQR])		220 (126, 382)	NA	189 (122, 276)	182 (81, 392)
Bilirubin µmol/L (median [IQR])		13 (9, 20)	NA	31 (15, 91)	8 (7, 28)

* In discovery cohort, chronic liver diseases included primary sclerosing cholangitis (*n* = 3), liver cirrhosis (*n* = 2), both primary sclerosing cholangitis and liver cirrhosis (*n* = 1), and hepatitis B (*n* = 2). In second-line cohort, chronic liver diseases included liver cirrhosis (*n* = 1), hepatitis B (*n* = 2), and hepatitis C (*n* = 1). Abbreviations: ALAT—alanine transaminase; ALP—alkaline phosphatase; ASAT—aspartate transaminase; dCC—distal cholangiocarcinoma; GBC—gallbladder cancer; Gem—gemcitabine; GemCapOx—gemcitabine + capecitabine + oxaliplatin; GemCapIri—gemcitabine + capecitabine + irinotecan; GemCis—gemcitabine + cisplatin; iCC—intrahepatic cholangiocarcinoma; IQR—interquartile range; mAB—anti-epidermal growth factor monoclonal antibody (cetuximab or panitumumab) or anti-vascular endothelial growth factor A monoclonal antibody (bevacizumab); OS—overall survival; pCC—perihilar cholangiocarcinoma.

**Table 2 cancers-15-01062-t002:** Performance of signatures in all cohorts.

Sig *	*n*	AUC Disc Det	AUC Disc Rep	AUC First-Line Val	AUC Surgery	AUC Second-Line
Set 1, Sig. 1	90	0.80 (0.67–0.92)	0.73 (0.63–0.83)	0.64 (0.54–0.75)	0.70 (0.54–0.86)	0.73 (0.58–0.87)
Set 1, Sig. 2	54	0.83 (0.71–0.94)	0.72 (0.61–0.82)	0.65 (0.55–0.76)	0.72 (0.57–0.88)	0.71 (0.56–0.86)
Set 1, Sig. 3	32	0.87 (0.76–0.97)	0.71 (0.61–0.81)	0.67 (0.57–0.77)	0.67 (0.50–0.84)	0.68 (0.51–0.84)
Set 1, Sig. 4	16	0.86 (0.75–0.97)	0.74 (0.63–0.84)	0.65 (0.55–0.75)	0.63 (0.44–0.82)	0.67 (0.50–0.84)
Set 1, Sig. 5	10	0.85 (0.73–0.97)	0.72 (0.62–0.82)	0.66 (0.55–0.76)	0.66 (0.49–0.84)	0.70 (0.54–0.86)
Set 1, Sig. 7	6	0.85 (0.72–0.97)	0.73 (0.63–0.83)	0.65 (0.55–0.76)	0.65 (0.48–0.83)	0.70 (0.54–0.86)
Set 1, Sig. 8	5	0.84 (0.71–0.97)	0.75 (0.65–0.84)	0.66 (0.56–0.77)	0.68 (0.51–0.85)	0.69 (0.53–0.86)
Set 1, Sig. 9	4	0.84 (0.71–0.97)	0.74 (0.64–0.84)	0.66 (0.56–0.76)	0.65 (0.46–0.83)	0.76 (0.62–0.89)
Set 1, Sig. 10	3	0.84 (0.71–0.97)	0.71 (0.60–0.81)	0.64 (0.54–0.74)	0.70 (0.52–0.88)	0.74 (0.58–0.90)
Set 1, Sig. 13	2	0.80 (0.67–0.93)	0.63 (0.52–0.75)	0.59 (0.49–0.70)	0.70 (0.53–0.87)	0.64 (0.48–0.80)
Set 1, Sig. 14	1	0.67 (0.51–0.84)	0.58 (0.46–0.69)	0.55 (0.44–0.66)	0.61 (0.44–0.79)	0.46 (0.29–0.64)
Set 2, Sig. 1	90	0.91 (0.81–1.00)	0.86 (0.77–0.96)	0.79 (0.68–0.90)	0.85 (0.71–0.99)	0.85 (0.71–0.98)
Set 2, Sig. 2	39	0.96 (0.91–1.00)	0.88 (0.80–0.96)	0.82 (0.70–0.93)	0.81 (0.64–0.98)	0.88 (0.77–0.99)
Set 2, Sig. 3	23	0.99 (0.98–1.00)	0.89 (0.81–0.97)	0.82 (0.71–0.93)	0.85 (0.71–0.98)	0.89 (0.78–0.99)
Set 2, Sig. 4	19	1.00 (1.00–1.00)	0.89 (0.80–0.97)	0.82 (0.71–0.93)	0.83 (0.67–0.98)	0.90 (0.80–0.99)
Set 2, Sig. 5	11	1.00 (1.00–1.00)	0.90 (0.82–0.98)	0.83 (0.72–0.93)	0.85 (0.71–0.98)	0.86 (0.74–0.98)
Set 2, Sig. 6	9	1.00 (1.00–1.00)	0.91 (0.83–0.99)	0.82 (0.71–0.93)	0.85 (0.71–0.98)	0.86 (0.75–0.98)
Set 2, Sig. 7	6	0.99 (0.98–1.00)	0.91 (0.83–0.98)	0.82 (0.72–0.93)	0.85 (0.71–0.98)	0.87 (0.75–0.99)
Set 2, Sig. 9	4	0.99 (0.98–1.00)	0.90 (0.83–0.98)	0.84 (0.74–0.95)	0.79 (0.62–0.95)	0.86 (0.71–1.00)
Set 2, Sig. 12	3	1.00 (1.00–1.00)	0.89 (0.80–0.98)	0.85 (0.74–0.95)	0.76 (0.56–0.96)	0.81 (0.61–1.00)
Set 2, Sig. 18	2	0.99 (0.98–1.00)	0.89 (0.80–0.98)	0.83 (0.72–0.94)	0.62 (0.30–0.94)	0.80 (0.60–1.00)
Set 2, Sig. 19	1	0.89 (0.78–1.00)	0.84 (0.73–0.94)	0.77 (0.63–0.91)	0.50 (0.18–0.82)	0.76 (0.53–1.00)
Set 3, Sig. 1	90	0.78 (0.54–1.00)	0.92 (0.83–1.00)	0.74 (0.55–0.93)	0.71 (0.47–0.96)	0.75 (0.45–1.00)
Set 3, Sig. 2	19	0.94 (0.83–1.00)	0.92 (0.84–1.00)	0.75 (0.57–0.94)	0.73 (0.50–0.96)	0.80 (0.53–1.00)
Set 3, Sig. 3	12	1.00 (1.00–1.00)	0.93 (0.84–1.00)	0.74 (0.55–0.93)	0.70 (0.45–0.95)	0.82 (0.55–1.00)
Set 3, Sig. 4	7	1.00 (1.00–1.00)	0.96 (0.91–1.00)	0.76 (0.57–0.95)	0.73 (0.49–0.97)	0.84 (0.62–1.00)
Set 3, Sig. 5	3	1.00 (1.00–1.00)	0.92 (0.83–1.00)	0.81 (0.65–0.97)	0.70 (0.39–1.00)	0.92 (0.78–1.00)
Set 3, Sig. 13	2	0.96 (0.88–1.00)	0.92 (0.83–1.00)	0.82 (0.67–0.98)	0.59 (0.18–1.00)	0.96 (0.87–1.00)

* Set 1—comparing patients with overall survival (OS) ≤ 12 months vs. OS > 12 months; Set 2—comparing OS ≤ 6 months vs. OS > 18 months; and Set 3—comparing OS ≤ 3 months vs. OS > 24 months. All values are presented with bootstrapped 95% confidence intervals in parentheses. Abbreviations: AUC—area under the receiver operating characteristic curve; Det—detection set; Disc—discovery cohort; *n*—number of proteins; Rep—replication set; Sig—signature; Val—validation cohort.

## Data Availability

The data that support the findings of this study are available from the corresponding author upon reasonable request.

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
