# Peer review of "Protein Signatures and Individual Circulating Proteins, including IL-6 and IL-15, Associated with Prognosis in Patients with Biliary Tract Cancer"

_cancers, 2023, doi:10.3390/cancers15041062_

Round 1

Reviewer 1 Report

In this study, the author has studied “Protein signatures and individual circulating proteins, including IL-6 and IL-15, associated with prognosis in patients with biliary tract cancer.” This is an engaging article with a robust approach that purposefully questions our knowledge of the subject. However, the presentation of the methodology is somewhat confusing, and the readability of the discussion could be improved. Addressing both these issues will make this interesting paper more impactful. The English language used in the manuscript needs some improvements as there are some punctuation and grammatical mistakes present throughout the manuscript.

Specific comments:

1.      Page 1, line 28: “The study included patients with BTC treated at three Danish hospitals.” It is suggested to add the number of patients for more clarity.

2.      The Abstract needs to be critically revised, please add some background knowledge and a clear objective.

3.      Please add more strong keywords.

4.      Please add more data in the first paragraph of the introduction and make it 10-15 lines long.

5.      Page 2, line 54-55: “The highest survival rate is achieved in patients where the cancer can be surgically 54 removed.” How much survival rate is estimated?

6.      Page 2: The whole introduction section looks general. Authors are advised to revise the introduction section carefully and add relevant data to support the problem statement and make a connection between each paragraph. There is no such information between proteins and BTC. The authors only discussed general information about BTC and its treatment. Overall, an introduction needs a major revision.

7.      Page 2: What is the research gap and novelty of the present study?

8.      Page 3, line 103-104: “prospective clinical studies between October 2008 and May 2021 at three Danish hospitals.” Please include the names of the hospitals.

9.      Page 3, line 132-133: “and centrifuged at 2300 g at 4ËšC for 132 10 minutes.” There is always a space between a value and a unit (4 ℃). Please also revise ‘minutes’ to ‘min.’

10.  What were the inclusion and exclusion criteria?

11.  The methodology section is not clear. The sample size and characteristics of patients are missing in this section.

12.  Page 5, line 217: “A total of 89 proteins were used in the analyses.” It is suggested to include all proteins as a supplementary file.

13.  Page 7, line 245-246: “The 10 proteins with lowest p-value were…” Did the performance was judged just based on the lowest p-value?

14.  Please add more data in the conclusion section. It should contain sufficient data because it will be indexed on multiple platforms after publication. That’s why it should contain sufficient data to present the idea of the study for better understanding.

15.  Authors are advised to proofread the manuscript to overcome grammatical mistakes.

16. It is suggested to add a list of abbreviations

Author Response

Thank you very much for a thorough and constructive peer review. We have used your comments to revise and improve the manuscript (please see point-to-point response). All changes from the prior version are highlighted. 

Besides changes recommended by you and the other reviewers, we have added reference 47 and a small discussion about the link between IL-15 and liver inflammation since it might explain our observations. We also updated the reference to our previous study (reference 28) that has been published in JHEP reports while the manuscript was under review. 

Point-to-point response

In this study, the author has studied “Protein signatures and individual circulating proteins, including IL-6 and IL-15, associated with prognosis in patients with biliary tract cancer.” This is an engaging article with a robust approach that purposefully questions our knowledge of the subject. However, the presentation of the methodology is somewhat confusing, and the readability of the discussion could be improved. Addressing both these issues will make this interesting paper more impactful. The English language used in the manuscript needs some improvements as there are some punctuation and grammatical mistakes present throughout the manuscript.

                      RESPONSE: We have edited the discussion and removed non-essential parts to increase readability. The method section has also been edited as advised. We have also performed additional English language revisions.

Specific comments:

  1. Page 1, line 28: “The study included patients with BTC treated at three Danish hospitals.” It is suggested to add the number of patients for more clarity.

                      RESPONSE: The total number of patients included in the study has been added. For consistency we have changed number of patients in each cohort, so it specifies number of patients included prior to QC. In the prior version, the abstract showed the number of patients after removing patients that failed protein analysis QC.

  1. The Abstract needs to be critically revised, please add some background knowledge and a clear objective.

                      RESPONSE: We have revised the manuscript and defined the objective more clearly. The abstract is already long (261 words), and it is difficult to add more background knowledge without removing essential methods and results.

  1. Please add more strong keywords.

                      RESPONSE: We have added two extra key words: “biliary tract cancer, interleukin”

  1. Please add more data in the first paragraph of the introduction and make it 10-15 lines long.

                      RESPONSE: We have added additional data to the paragraph and the subsequent paragraph.

  1. Page 2, line 54-55: “The highest survival rate is achieved in patients where the cancer can be surgically 54 removed.” How much survival rate is estimated?

                      RESPONSE: We have changed the sentence and added data concerning 5-year survival after surgery and a additional reference (reference 6)

  1. Page 2: The whole introduction section looks general. Authors are advised to revise the introduction section carefully and add relevant data to support the problem statement and make a connection between each paragraph. There is no such information between proteins and BTC. The authors only discussed general information about BTC and its treatment. Overall, an introduction needs a major revision.

                      RESPONSE: We have performed a major revision of the introduction section as suggested.

  1. Page 2: What is the research gap and novelty of the present study?

                      RESPONSE: As part of the revision of the introduction section we have highlighted the research gaps, including lack of high accuracy prognostic markers (page 2, line 86-87) and the potential of circulating protein and the lack of studies testing different proteins (page 2, line 88-101).

  1. Page 3, line 103-104: “prospective clinical studies between October 2008 and May 2021 at three Danish hospitals.” Please include the names of the hospitals.

                      RESPONSE: Hospital names have been added as suggested.

  1. Page 3, line 132-133: “and centrifuged at 2300 g at 4ËšC for 132 10 minutes.” There is always a space between a value and a unit (4 ℃). Please also revise ‘minutes’ to ‘min.’

                      RESPONSE: We have revised the section as suggested.

  1. What were the inclusion and exclusion criteria?

                      RESPONSE: We have revised the Patients section so exclusion and inclusion criteria are more clearly described (Page 3, line 133-142).

  1. The methodology section is not clear. The sample size and characteristics of patients are missing in this section.

                      RESPONSE: We specified number of included patients in patients´ section. Furthermore, we have moved Figure 1 to the methods section to clarify patient selection. Baseline characteristics are provided in Table 1.        

  1. Page 5, line 217: “A total of 89 proteins were used in the analyses.” It is suggested to include all proteins as a supplementary file.

                      RESPONSE: A complete list of proteins are provided in Table S1. The three proteins were removed due to more than 90% of samples being below limit of detection (LOD). It is advised by the manufacturer not to use these proteins in the analysis since results are not reliable when levels are below LOD. We have therefore not performed the analysis for the last three proteins.

  1. Page 7, line 245-246: “The 10 proteins with lowest p-value were…” Did the performance was judged just based on the lowest p-value?

                      REPONSE: The performance was only judged on the p-value as decided prior to analysis in our analysis plan. Of cause, other parameters could have been used, but for simplicity we had decided to use the p-value.

  1. Please add more data in the conclusion section. It should contain sufficient data because it will be indexed on multiple platforms after publication. That’s why it should contain sufficient data to present the idea of the study for better understanding.

                      RESPONSE: We have added more data in the conclusion section as suggested.

  1. Authors are advised to proofread the manuscript to overcome grammatical mistakes.

                      RESPONSE: We have checked for and corrected grammatical mistakes.

  1. It is suggested to add a list of abbreviations

                      RESPONSE: As we understand from the author instruction of Cancers, a list of abbreviations is not usually recommended, and we have therefor not added it.

Reviewer 2 Report

The manuscript is well written and includes the detailed analysis.

Author Response

Thank you for reviewing the manuscript and kind words

Reviewer 3 Report

In this study, Christensen et al. examined the link between a number of cancer-related cytokines and proteins and the prognosis of biliary tract carcinoma (BTC). In this investigation, BTC patients treated at three different Danish hospitals had their plasma protein levels measured. The researchers found that elevated levels of the cytokines interleukin (IL)-6, IL-15, mucin 16, hepatocyte growth factor, programmed cell death ligand 1, and placental growth factor were linked to lower patient overall survival.

comments

1-There are several previous studies shows prognostic role of the IL-6 in CCA or BTC patients, which limits the novelty of this study.

  • DOI: 10.1111/j.1572-0241.2007.01403.x

Doi.org/10.1158/0008-5472.CAN-06-2130

2- Were, the level of IL-6 and Il-15 also changed with mutations (IDH, EGFR or FGF) in the patients

3- Did authors also checked the difference in these biomarkers with the different stages of the bilebuct cancer?

Author Response

Thank you very much for peer-reviewing our manuscript and insightful comments. We have used your comments to revise and improve the manuscript (please see point-to-point response).

Point-to-point response:

In this study, Christensen et al. examined the link between a number of cancer-related cytokines and proteins and the prognosis of biliary tract carcinoma (BTC). In this investigation, BTC patients treated at three different Danish hospitals had their plasma protein levels measured. The researchers found that elevated levels of the cytokines interleukin (IL)-6, IL-15, mucin 16, hepatocyte growth factor, programmed cell death ligand 1, and placental growth factor were linked to lower patient overall survival.

comments

1-There are several previous studies shows prognostic role of the IL-6 in CCA or BTC patients, which limits the novelty of this study.

  • DOI: 10.1111/j.1572-0241.2007.01403.x

Doi.org/10.1158/0008-5472.CAN-06-2130

                      RESPONSE: We agree that IL-6 has previously been associated with survival in BTC. This is also mentioned in the discussion section. However, the present study adds information about the prognostic ability of IL-6 as compared to the other 88 proteins tested and the correlation between these. This information is valuable in the understanding of the general blood protein landscape.  

2- Were, the level of IL-6 and Il-15 also changed with mutations (IDH, EGFR or FGF) in the patients.

RESPONSE: It would of cause be interesting to compare the protein level with mutations. Unfortunately, most of the patients were treated before routine mutation analysis was performed at our clinic, and we do not have sufficient data to check how mutations impact the association. We have therefore added it as a limitation to the study. Furthermore, we hope to be able to test this in an ongoing trial (Eudract 2018-004826-27) where we test for mutational status using Foundation One Liquid as well as target tissue panels. The trial is expected to finish within a couple of years.

3- Did authors also checked the difference in these biomarkers with the different stages of the bile duct cancer?

                      RESPONSE: This is an interesting proposal. As suggested, we have compared protein level to stage and cohort (surgery, first line, or second line). Four lines have been added in results section 3.3 (page 12, lines 390-393) as well as a supplementary figure S5.

Round 2

Reviewer 1 Report

The authors have carefully addressed all the comments. So, the manuscript should be accepted in its present form.

Reviewer 3 Report

I am pleased to see that the revisions have addressed the concerns that I raised in my review.